# DeepInteraction: 3D Object Detection via Modality Interaction

**Zeyu Yang**[1] **Jiaqi Chen**[1] **Zhenwei Miao**[2] **Wei Li**[3] **Xiatian Zhu**[4] **Li Zhang**[1*]

[1]Fudan University   [2]Alibaba DAMO Academy   [3]S-Lab, NTU   [4]University of Surrey

https://github.com/fudan-zvg/DeepInteraction

## Abstract

Existing top-performance 3D object detectors typically rely on the multi-modal fusion strategy. This design is however fundamentally restricted due to overlooking the modality-specific useful information and finally hampering the model performance. To address this limitation, in this work we introduce a novel modality interaction strategy where individual per-modality representations are learned and maintained throughout for enabling their unique characteristics to be exploited during object detection. To realize this proposed strategy, we design a `DeepInteraction` architecture characterized by a multi-modal representational interaction encoder and a multi-modal predictive interaction decoder. Experiments on the large-scale nuScenes dataset show that our proposed method surpasses all prior arts often by a large margin. Crucially, our method is ranked at the `first` position at the highly competitive nuScenes object detection leaderboard.

## 1   Introduction

3D object detection is critical for autonomous driving by localizing and recognizing decision-sensitive objects in a 3D world. For reliable object detection, LiDAR and camera sensors have been simultaneously deployed to provide *point clouds* and *RGB images* for more stronger perception. The two modalities exhibit naturally strong complementary effects due to their different perceiving characteristics. Point clouds offer necessary localization and geometry information at low resolution, whilst images give rich appearance information at high resolution. Therefore, *information fusion* across modalities becomes particularly crucial for strong 3D object detection performance.

Existing multi-modal 3D objection detection methods typically adopt a ***modality fusion*** strategy (Figure 1(a)) by combining individual per-modality representations into a *single* hybrid representation. For example, PointPainting [38] and its variants [39, 46, 43] aggregate category scores or semantic features from the image space into the 3D point cloud space. AutoAlign [10] and VFF [23] similarly integrate image representations into the 3D grid space. Latest alternatives [24, 30, 26] merge the image and point cloud features into a joint bird's-eye view (BEV) representation. This fusion approach is, however, structurally restricted due to its intrinsic limitation of potentially dropping off a large fraction of modality-specific representational strengths due to largely imperfect information fusion into a unified representation.

To overcome the aforementioned limitations, in this work a novel ***modality interaction*** strategy (Figure 1(b)) for multi-modal 3D object detection is introduced. Our key idea is that, instead of deriving a fused single representation, we learn and maintain two modality-specific representations throughout to enable inter-modality interaction so that both information exchange and modality-specific strengths can be achieved spontaneously. Our strategy is implemented by formulating a

---

*Li Zhang (lizhangfd@fudan.edu.cn) is the corresponding author with School of Data Science, Fudan University.

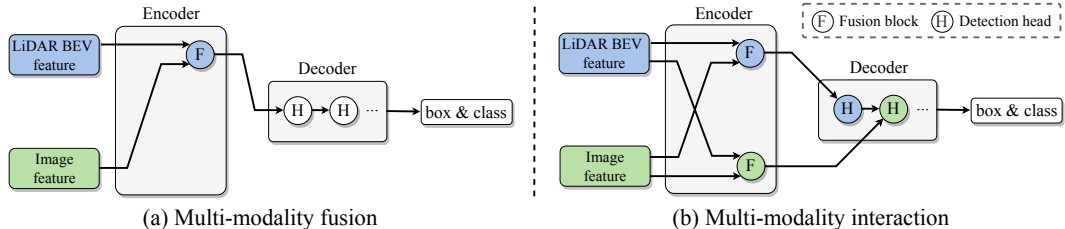

Figure 1: **Schematic strategy comparison**. **(a)** Existing multi-modality fusion based 3D detection: Fusing individual per-modality representations into a single hybrid representation and from which the detection results are further decoded. **(b)** Our multi-modality interaction based 3D detection: Maintaining two modality-specific representations throughout the whole pipeline with both *representational interaction* in the encoder and *predictive interaction* in the decoder.

`DeepInteraction` architecture. It starts by mapping 3D point clouds and 2D multi-view images into LiDAR BEV feature and image perspective feature with two separate feature backbones in parallel. Subsequently, an encoder interacts the two features for progressive information exchange and representation learning in a *bilateral* manner. To fully exploit per-modality representations, a decoder/head is further designed to conduct multi-modal predictive interaction in a cascaded manner.

Our **contributions** are summarized as follows: **(i)** We propose a novel *modality interaction* strategy for multi-modal 3D object detection, with the aim to resolve a fundamental limitation of previous *modality fusion* strategy in dropping the unique perception strengths per modality. **(ii)** To implement our proposed strategy, we formulate a `DeepInteraction` architecture with a multi-modal representational interaction encoder and a multi-modal predictive interaction decoder. **(iii)** Extensive experiments on the nuScenes dataset show that our `DeepInteraction` yields new state of the art for multi-modality 3D object detection and achieves the first position at the highly competitive nuScenes leaderboard.

## 2  Related work

**3D object detection with single modality**  Automated driving vehicles are generally equipped with both LiDAR and multiple surround-view cameras. But many previous methods perform 3D object detection by exploiting data captured from only a single form of sensor. For camera-based 3D object detection, since depth information is not directly accessible from RGB images, some previous works [17, 40, 37] lift 2D features into a 3D space by conducting depth estimation, followed by performing object detection in the 3D space. Another line of works [41, 28, 25, 31, 19] resort to the detection Transformer [5] architecture. They leverage 3D object queries and 3D-2D correspondence to incorporate 3D computation into the detection pipelines. Despite the rapid progress of camera-based approaches, the state-of-the-art of 3D object detection is still dominated by LiDAR-based methods. Most of LiDAR-based detectors quantify point clouds into regular grid structures such as voxels [47, 44], pillars [22] or range images [2, 14, 6] before processing them. Due to the sampling characteristics of LiDAR, these grids are naturally sparse and hence fit the Transformer design. So a number of approaches [32, 13] have applied the Transformer for point cloud feature extraction. Differently, several methods use the Transformer decoder or its variants as their detection head [1, 42]. 3DETR [33] adopts a complete Transformer encoder-decoder architecture with less priori in design. Due to intrinsic limitations with either sensor, these methods are largely limited in performance.

**Multi-modality fusion for 3D object detection**  Leveraging the perception data from both camera and LiDAR sensors usually leads to improved performance. This has emerged as a promising direction. Existing 3D detection methods typically perform multi-modal fusion at one of the three stages: raw input, intermediate feature, and object proposal. For example, PointPainting [38] is the pioneering input fusion method [39, 18, 46]. The main idea is to decorate the 3D point clouds with the category scores or semantic features from the 2D instance segmentation network. Whilst 4D-Net [35] placed the fusion module in the point cloud feature extractor for allowing the point cloud features to dynamically attend to the image features. ImVoteNet [36] injects visual information into a set of 3D seed points abstracted from raw point clouds. The proposal based fusion methods [21, 7]

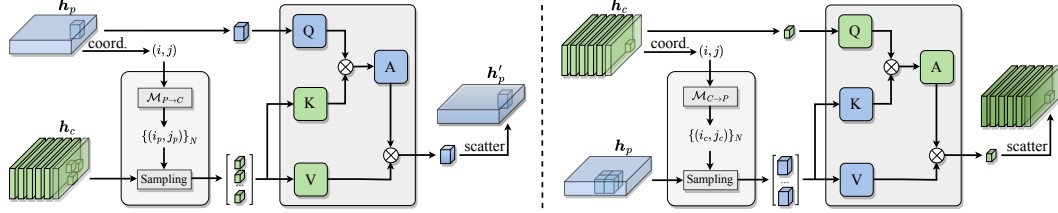

(a) Representational interaction from image to LiDAR      (b) Representational interaction from LiDAR to image

Figure 2: Illustration of the multi-modal representational interactions. Given two modality-specific representations, the image-to-LiDAR feature interaction **(a)** spread the visual signal in the image representation to the LiDAR BEV representation, and the LiDAR-to-image feature interaction **(b)** takes cross-modal relative contexts from LiDAR representation to enhance the image representations.

keep the feature extraction of two modalities independently and aggregate multi-modal features via proposals or queries at the detection head. The first two categories of methods take a unilateral fusion strategy with bias to 3D LiDAR modality due to the superiority of point clouds in distance and spatial perception. Instead, the last category fully ignores the intrinsic association between the two modalities in representation. As a result, all above previous methods fail to fully exploit both modalities, in particular their strong complementary nature. Besides, a couple of concurrent works have explored fusion of the two modalities in a shared representation space [30, 26]. They conduct view transformation in the same way [34] as in the camera-only approach. This design is however less effective in exploiting the spatial cues of point clouds during view transformation, potentially compromising the quality of camera BEV representation. This gives rise to an extra need of calibrating such misalignment in network capacity.

In this work, we address the aforementioned limitations in all previous solutions with a novel multi-modal interaction strategy. The key insight behind our approach is that we maintain two modality-specific feature representations and conduct *representational* and *predictive* interaction for maximally exploring their complementary benefits whilst preserving their respective strengths.

## 3    Method

We present a novel modality interaction framework, dubbed `DeepInteraction`, for multi-modal (3D point clouds and 2D multi-camera images) 3D object detection. In contrast to all prior arts, we learn two representations specific for 3D LiDAR and 2D image modalities respectively, whilst conducting multi-modal interaction through both model encoding and decoding. An overview of `DeepInteraction` is shown in Figure 1(b). It consists of two main components: An encoder with multi-modal representational interaction (Section 3.1), and a decoder with multi-modal predictive interaction (Section 3.2).

### 3.1    Encoder: Multi-modal representational interaction

Unlike conventional modality fusion strategy that often aggregates multi-modal inputs into a hybrid feature map, ***individual per-modality representations*** are learned and maintained via *multi-modal representational interaction* within our encoder. Specifically, our encoder is formulated as a *multi-input-multi-output* (MIMO) structure: Taking as input two modality-specific scene representations which are independently extracted by LiDAR and image backbones, and producing two refined representations as output. Overall, it is composed by stacking multiple encoder layers each with *(I) multi-modal representational interaction* (MMRI), *(II) intra-modal representational learning* (IML), and *(III) representational integration*.

**(I) Multi-modal representational interaction (MMRI)**    Each encoder layer takes the representations of two modalities, *i.e.*, the image perspective representation $\boldsymbol{h}_c$ and the LiDAR BEV representation $\boldsymbol{h}_p$, as input. Our multi-modal representational interaction aims to exchange the neighboring context in a bilateral cross-modal manner, as shown in Figure 2. It consists of two steps:

***(i) Cross-modal correspondence mapping and sampling***  To define cross-modality adjacency, we first need to build the pixel-to-pixel(s) correspondence between the representations $\boldsymbol{h}_p$ and $\boldsymbol{h}_c$. To that end, we construct dense mappings between the image coordinate frame $c$ and the BEV coordinate frame $p$ ($\mathcal{M}_{p\rightarrow c}$ and $\mathcal{M}_{c\rightarrow p}$).

*From image to LiDAR BEV coordinate frame* $\mathcal{M}_{c\rightarrow p}: \mathbb{R}^2 \rightarrow 2^{\mathbb{R}^2}$ (Figure 2(a)): We first project each point $(x, y, z)$ in 3D point cloud to multi-camera images to form a sparse depth map $\boldsymbol{d}_{sparse}$, followed by depth completion [20] leading to a dense depth map $\boldsymbol{d}_{dense}$. We further utilize $\boldsymbol{d}_{dense}$ to back-project each pixel in the image space into the 3D point space. This results in the corresponding 3D coordinate $(x, y, z)$, given an image pixel $(i, j)$ with depth $\boldsymbol{d}_{dense}^{[i,j]}$. Next, $(x, y)$ is used to locate the corresponding BEV coordinate $(i_p, j_p)$. We denote the above mapping as $T(i, j) = (i_p, j_p)$. We obtain this correspondence via $(2k + 1) \times (2k + 1)$ sized neighbor sampling as $\mathcal{M}_{c\rightarrow p}(i, j) = \{T(i + \Delta i, j + \Delta j) | \Delta i,\ \Delta j \in [-k, +k]\}$.

*From LiDAR BEV to image coordinate frame* $\mathcal{M}_{p\rightarrow c}: \mathbb{R}^2 \rightarrow 2^{\mathbb{R}^2}$ (Figure 2(b)): Given a coordinate $(i_p, j_p)$ in BEV, we first obtain the LiDAR points $\{(x, y, z)\}$ within the pillar corresponding to $(i_p, j_p)$. Then we project these 3D points into camera image coordinate frame $\{(i, j)\}$ according to the camera intrinsics and extrinsics. This correspondence is obtained as: $\mathcal{M}_{p\rightarrow c}(i_p, j_p) = \{(i, j)\}$.

***(ii) Attention-based feature interaction***  For an image feature point as query $\boldsymbol{q} = \boldsymbol{h}_c^{[i_c, j_c]}$, its cross-modality neighbors $\mathcal{N}_q$, denoted as $\mathcal{N}_q = \boldsymbol{h}_p^{[\mathcal{M}_{c\rightarrow p}(i_c, j_c)]}$, are used as the `key` $\boldsymbol{k}$ and `value` $\boldsymbol{v}$ for cross-attention learning:

$$f_{\phi_{c\rightarrow p}}(\boldsymbol{h}_c, \boldsymbol{h}_p)^{[i,j]} = \sum_{\boldsymbol{k}, \boldsymbol{v} \in \mathcal{N}_q} \text{softmax}\left(\frac{\boldsymbol{q}\boldsymbol{k}}{\sqrt{d}}\right)\boldsymbol{v}, \tag{1}$$

where $\boldsymbol{h}^{[i,j]}$ denotes indexing the element at location $(i, j)$ on the 2D representation $\boldsymbol{h}$. This is *image-to-LiDAR representational interaction*.

The other way around, given a LiDAR BEV feature point as query $\boldsymbol{q} = \boldsymbol{h}_p^{[i_p, j_p]}$, we similarly obtain its cross-modality neighbors as $\mathcal{N}_q = \boldsymbol{h}_c^{[\mathcal{M}_{p\rightarrow c}(i_p, j_p)]}$. The same process (Eq. (1)) is applied for realizing *LiDAR-to-image representational interaction* $f_{\phi_{p\rightarrow c}}(\boldsymbol{h}_c, \boldsymbol{h}_p)$.

**(II) Intra-modal representational learning (IML)**  Concurrently, we conduct intra-modal representational learning complementary to multi-modal interaction. The same local attention as defined in Eq. (1) is consistently applied. For either modality, we use a $k \times k$ grid neighborhood as the `key` and `value`. Formally, we denote $f_{\phi_{c\rightarrow c}}(\boldsymbol{h}_c)$ for image representation and $f_{\phi_{p\rightarrow p}}(\boldsymbol{h}_p)$ for LiDAR representation.

**(III) Representational integration**  Each layer ends up by integrating the two outputs per modality:

$$\begin{aligned}
\boldsymbol{h}_p' &= \text{FFN}(\text{Concat}(\text{FFN}(\text{Concat}(\boldsymbol{h}_p^{p\rightarrow p},\ \boldsymbol{h}_p^{c\rightarrow p})),\ \boldsymbol{h}_p)), \\
\boldsymbol{h}_c' &= \text{FFN}(\text{Concat}(\text{FFN}(\text{Concat}(\boldsymbol{h}_c^{c\rightarrow c},\ \boldsymbol{h}_c^{p\rightarrow c})),\ \boldsymbol{h}_c)),
\end{aligned} \tag{2}$$

where FFN specifies a feed-forward network, and Concat denotes element-wise concatenation.

## 3.2  Decoder: Multi-modal predictive interaction

Beyond considering the multi-modal interaction at the representation-level, we further introduce a decoder with *multi-modal predictive interaction* (MMPI) to maximize the complementary effects in prediction. As depicted in Figure 3(a), our core idea is to enhance the 3D object detection of one modality conditioned on the other modality. In particular, the decoder is built by stacking multiple *multi-modal predictive interaction layers*, within which *predictive interactions* are formulated in an alternative and progressive manner. Similar to the decoder of DETR [5], we cast the 3D object detection as a set prediction problem. Here, we define a set of $N$ object queries $\{\boldsymbol{Q}_n\}_{n=1}^N$ and the resulting $N$ object predictions $\{(\boldsymbol{b}_n, \boldsymbol{c}_n)\}_{n=1}^N$, where $\boldsymbol{b}_n$ and $\boldsymbol{c}_n$ denote the predicted bounding box and category for the $n$-th prediction.

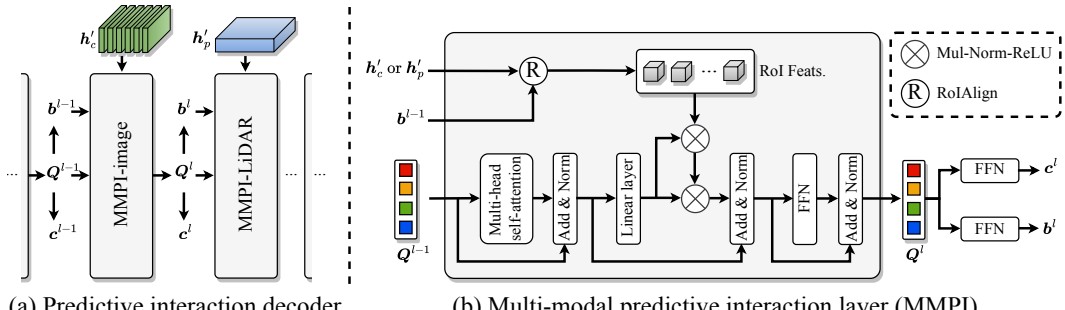

(a) Predictive interaction decoder          (b) Multi-modal predictive interaction layer (MMPI)

Figure 3: Illustration of our multi-modal predictive interaction. Our predictive interaction decoder **(a)** generates predictions via **(b)** progressively interacting with two modality-specific representations.

**Multi-modal predictive interaction layer (MMPI)** For the $l$-th decoding layer, the set prediction is computed by taking the object queries $\left\{Q_n^{(l-1)}\right\}_{n=1}^N$ and the bounding box predictions $\left\{b_n^{(l-1)}\right\}_{n=1}^N$ from previous layer as inputs and enabling interaction with the intensified image $h'_p$ or LiDAR $h'_c$ representations ($h'_c$ if $l$ is odd, $h'_p$ if $l$ is even). We formulate the multi-modal predictive interaction layer (Figure 3(b)) for specific modality as follows:

**(I) Multi-modal predictive interaction on image representation (MMPI-image)** Taking as input 3D object proposals $\left\{b_n^{(l-1)}\right\}_{n=1}^N$ and object queries $\left\{Q_n^{(l-1)}\right\}_{n=1}^N$ generated by the previous layer, this layer leverages the image representation $h'_c$ for further prediction refinement. To integrate the previous predictions $\left\{b_n^{(l-1)}\right\}_{n=1}^N$, we first extract $N$ Region of Interest (RoI) [15] features $\{R_n\}_{n=1}^N$ from the image representation $h'_c$, where $R_n \in \mathbb{R}^{S \times S \times C}$ is the extracted RoI feature for the $n$-th query, $(S \times S)$ is RoI size, and $C$ is the number of channels of RoI features. Specifically, for each 3D bounding box, we project it onto image representation $h'_c$ to get the 2D convex polygon and take the minimum *axis-aligned* circumscribed rectangle. We then design a multi-modal predictive interaction operator that maps $\left\{Q_n^{(l-1)}\right\}_{n=1}^N$ into the parameters of a series of $1 \times 1$ convolutions and then applies them consecutively to $\{R_n\}_{n=1}^N$; The resulted interactive representation is further used to obtain the updated object query $\left\{Q_n^l\right\}_{n=1}^N$.

**(II) Multi-modal predictive interaction on LiDAR representation (MMPI-LiDAR)** This layer shares the same design as the above except that it takes as input LiDAR representation instead. With regards to the RoI for LiDAR representation, we project the 3D bounding boxes from previous layer to the LiDAR BEV representation $h'_p$ and take the minimum *axis-aligned* rectangle. It is worth mentioning that due to the scale of objects in autonomous driving scenarios is usually tiny in the BEV coordinate frame, we enlarge the scale of the 3D bounding box by 2 times. The shape of RoI features cropped from the LiDAR BEV representation $h'_p$ is also set to be $S \times S \times C$. Here $C$ is the $C$ is the number of channels of RoI features as well as the height of BEV representation. The multi-modal predictive interaction layer on LiDAR representation is stacked on the above image counterpart.

For object detection, a feed-forward network is appended on the $\left\{Q_n^l\right\}_{n=1}^N$ for each multi-modal predictive interaction layer to infer the locations, dimensions, orientations and velocities. During training, the matching cost and loss function as [1] are applied.

## 4 Experiments

### 4.1 Experimental setup

**Dataset** We evaluate our approach on the nuScenes dataset [3]. It provides point clouds from 32-beam LiDAR and images with a resolution of $1600 \times 900$ from 6 surrounding cameras. The total of 1000 scenes, where each sequence is roughly 20 seconds long and annotated every 0.5 second, is

Table 1: Comparison with state-of-the-art methods on the nuScenes `test` set. `Metrics`: mAP(%), NDS(%). 'L' and 'C' represent LiDAR and camera, respectively. † denotes test-time augmentation is used. § denotes that test-time augmentation and model ensemble both are applied for testing.

| Method | Modality | Backbones | | validation | | test | |
| | | Image | LiDAR | mAP↑ | NDS↑ | mAP↑ | NDS↑ |
|---|---|---|---|---|---|---|---|
| BEVDet4D [17] | C | Swin-Base | - | 42.1 | 54.5 | 45.1 | 56.9 |
| BEVFormer [25] | C | V99 | - | - | - | 48.1 | 56.9 |
| Ego3RT [31] | C | V99 | - | 47.8 | 53.4 | 42.5 | 47.9 |
| PolarFormer [19] | C | V99 | - | 50.0 | 56.2 | 49.3 | 57.2 |
| CenterPoint [45] | L | - | VoxelNet | 59.6 | 66.8 | 60.3 | 67.3 |
| Focals Conv [8] | L | - | VoxelNet-FocalsConv | 61.2 | 68.1 | 63.8 | 70.0 |
| Transfusion-L [1] | L | - | VoxelNet | 65.1 | 70.1 | 65.5 | 70.2 |
| LargeKernel3D [9] | L | - | VoxelNet-LargeKernel3D | 63.3 | 69.1 | 65.3 | 70.5 |
| FUTR3D [7] | L+C | R101 | VoxelNet | 64.5 | 68.3 | - | - |
| PointAugmenting [39]† | L+C | DLA34 | VoxelNet | - | - | 66.8 | 71.0 |
| MVP [46] | L+C | DLA34 | VoxelNet | 67.1 | 70.8 | 66.4 | 70.5 |
| AutoAlignV2 [10] | L+C | CSPNet | VoxelNet | 67.1 | 71.2 | 68.4 | 72.4 |
| TransFusion [1] | L+C | R50 | VoxelNet | 67.5 | 71.3 | 68.9 | 71.6 |
| BEVFusion [26] | L+C | Swin-Tiny | VoxelNet | 67.9 | 71.0 | 69.2 | 71.8 |
| BEVFusion [30] | L+C | Swin-Tiny | VoxelNet | 68.5 | 71.4 | 70.2 | 72.9 |
| **DeepInteraction-base** | L+C | R50 | VoxelNet | **69.9** | **72.6** | **70.8** | **73.4** |
| Focals Conv-F [8]† | L+C | R50 | VoxelNet-FocalsConv | 67.1 | 71.5 | 70.1 | 73.6 |
| LargeKernel3D-F [9]† | L+C | R50 | VoxelNet-LargeKernel | - | - | 71.1 | 74.2 |
| **DeepInteraction-large†** | L+C | Swin-Tiny | VoxelNet | **72.6** | **74.4** | **74.1** | **75.5** |
| BEVFusion-e [30]§ | L+C | Swin-Tiny | VoxelNet | 73.7 | 74.9 | 75.0 | 76.1 |
| **DeepInteraction-e§** | L+C | Swin-Tiny | VoxelNet | **73.9** | **75.0** | **75.6** | **76.3** |

officially split into `train/val/test` set with 700/150/150 scenes. For the 3D object detection task, 1.4M objects in scenes are annotated with 3D bounding boxes and classified into 10 categories: car, truck, bus, trailer, construction vehicle, pedestrian, motorcycle, bicycle, barrier, and traffic cone.

**Metric**   Mean average precision (mAP) [12] and nuScenes detection score (NDS) [3] are used as the evaluation metric of 3D detection performance. The final mAP is computed by averaging over the distance thresholds of 0.5m, 1m, 2m, 4m across 10 classes. NDS is a weighted average of mAP and other attribute metrics, including translation, scale, orientation, velocity, and other box attributes.

### 4.2   Implementation details

Our implementation is based on the public code base *mmdetection3d* [11]. For the image branch backbone, we use a simple ***ResNet-50*** [16] and initialize it from the instance segmentation model *Cascade Mask R-CNN* [4] pretrained on COCO [27] and then nuImage [3], which is same as Transfusion [1]. To save the computation cost, we rescale the input image to 1/2 of its original size before feeding into the network, and *freeze* the weights of image branch during training. The voxel size is set to $(0.075m, 0.075m, 0.2m)$, and the detection range is set to $[-54m, 54m]$ for $X$ and $Y$ axis and $[-5m, 3m]$ for $Z$ axis. The representational interaction encoder is composed by stacking two representational interaction layers. For the multi-modal predictive interaction decoder, we use 5 cascaded decoder layers. We set the query number to 200 for training and testing and use the same query initialization method as Transfusion [1]. The above configuration is termed `DeepInteraction-base`.

We also adopt another two widely used settings for online submission, *i.e.*, test-time augmentation (TTA) and model ensemble. In the following, we refer to the two settings as `DeepInteraction-large` and `DeepInteraction-e` respectively. In particular, `DeepInteraction-large` uses Swin-Tiny [29] as image backbone, and doubles the number of channel for each convolution block in LiDAR backbone. The voxel size of `DeepInteraction-large`

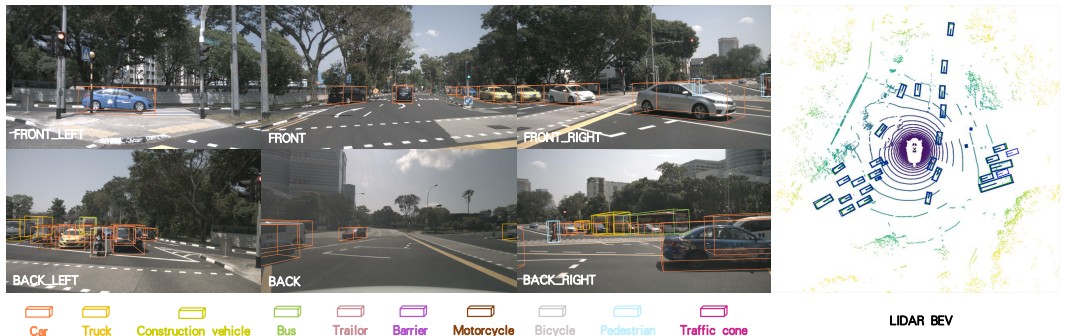

Car Truck Construction vehicle Bus Trailer Barrier Motorcycle Bicycle Pedestrian Traffic cone LIDAR BEV

Figure 4: Qualitative results on nuScenes `val` set. In LiDAR BEV (right), green boxes are the ground-truth and blue boxes are the predictions. Best viewed when zooming in.

is set to $[0.5m, 0.5m, 0.2m]$. Following the common practice, we use double flipping and rotation with yaw angles $[0°, ±6.25°, ±12.5°]$ for test-time augmentation. `DeepInteraction-e` ensembles multiple DeepInteraction-large models with input LiDAR BEV grid size between $[0.5m, 0.5m]$ and $[1.5m, 1.5m]$.

For data augmentation, following TransFusion [1] we adopt random rotation with a range of $r \in [-\pi/4, \pi/4]$, random scaling with a factor of $r \in [0.9, 1.1]$, random translation with standard deviation 0.5 in three axis, and random horizontal flipping. We also use the class-balanced re-sampling in CBGS [48] to balance the class distribution for nuScenes. Following [1], we adopt a two stage training recipe. We take TransFusion-L [1] as our `LiDAR-only baseline`. We use Adam optimizer with one-cycle learning rate policy, with max learning rate $1 \times 10^{-3}$, weight decay 0.01 and momentum 0.85 to 0.95, following CBGS [48]. Our LiDAR-only baseline is trained for 20 epochs and LiDAR-image fusion for 6 epochs with batch size of 16 using 8 NVIDIA V100 GPUs.

### 4.3 Comparison to the state of the art

**Performance** We compare with state-of-the-art alternatives on the nuScenes test set. As shown in Table 1, `DeepInteraction` achieves new state-of-the-art performance under all settings. The base variant without TTA and model ensemble, `DeepInteraction-base`, **with a simple ResNet-50 image backbone**, surpasses all the prior arts as well as the concurrent work BEVFusion [30] even with a Swin-Tiny image backbone. `DeepInteraction-large` beats the closest rival LargeKernel3D-F [9] with the same TTA and test time augmentation (single model) by a considerable margin. Our ensemble version `DeepInteraction-e` achieves the `first` rank among all the solutions on the nuScenes leaderboard. These results verify the performance advantages of our multi-modal interaction approach. Per-category results are shown in Appendix A.1. Qualitative results are provided in Figure 4 and Appendix A.4.

**Run time** We compare inference speed tested on NVIDIA V100, A6000 GPUs and A100 separately. As shown in Table 2, our method achieves the best performance while running faster than alternative painting-based [39] and query-based [7] fusion approaches. This validates superior trade-off between detection performance and inference speed achieved by our method. As found in [39], feature extraction for multi-view high resolution camera images contributes the most of overall latency in a multi-modal 3D detector. Indeed, from Table 3(c) we observe that increasing the number of decoder layers only brings negligible extra latency, which concurs with the same conclusion.

### 4.4 Ablations on the decoder

**Multi-modal predictive interaction layer vs. DETR [5] decoder layer** In Table 3(a) we evaluate the design of decoder layer by comparing our multi-modal predictive interaction (MMPI) with DETR [5] decoder layer. Note the DETR decoder layer means the conventional Transformer deocder layer is used to aggregate multi-modal information same as in Transfusion [1]. We further test a mixing design: using vanilla DETR decoder layer for aggregating features in LiDAR representation and our MMPI for aggregating features in image representation (second row). It is evident that MMPI

Table 2: Run time comparison. If not specified, the performance and efficiency are evaluated on the nuScenes `val` set.

| Method | mAP(%)↑ | NDS(%)↑ | FPS(A100)↑ | FPS(A6000)↑ | FPS(V100)↑ |
|---|---|---|---|---|---|
| PointAugmenting [39] | 66.8 (Test) | 71.0 (Test) | 1.4 | 2.8 | 2.3 |
| FUTR3D [7] | 64.2 | 68.0 | 4.5 | 2.3 | 1.8 |
| Transfusion [1] | 67.5 | 71.3 | **6.2** | **5.5** | **3.8** |
| **DeepInteraction** | **69.9** | **72.6** | 4.9 | 3.1 | 2.6 |

Table 3: Ablation studies on the decoder. The mAP and NDS are evaluated on the nuScenes `val` set.

| LiDAR | Image | mAP | NDS |
|---|---|---|---|
| DETR [5] | DETR [5] | 68.6 | 71.6 |
| DETR [5] | MMPI | 69.3 | 72.1 |
| MMPI | MMPI | **69.9** | **72.6** |

(a) The type of decoder layer.

| Modality | mAP | NDS |
|---|---|---|
| Fully LiDAR | 69.2 | 72.2 |
| LiDAR-image alternating | **69.9** | **72.6** |

(b) Single vs. multiple representations.

| # of decoder layers | mAP | NDS | FPS↑ |
|---|---|---|---|
| 1 (LiDAR-only) | 65.1 | 70.1 | 8.7 |
| 2 | 69.5 | 72.3 | 2.8 |
| 3 | 69.7 | 72.5 | 2.7 |
| 4 | 69.8 | 72.5 | 2.7 |
| 5 | **69.9** | **72.6** | 2.6 |
| 6 | 69.7 | 72.1 | 2.5 |

(c) The number of decoder layers.

| Train | Inference | mAP | NDS |
|---|---|---|---|
| | 200 | 69.9 | 72.6 |
| 200 | 300 | **70.1** | **72.7** |
| | 400 | 70.0 | 72.6 |
| | 200 | 69.7 | 72.5 |
| 300 | 300 | 69.9 | 72.6 |
| | 400 | 70.0 | 72.6 |

(d) The number of queries.

is significantly superior over DETR by improving 1.3% mAP and 1.0% NDS, with combinational flexibility in design.

**How many representations/modalities?** In Table 3(b), we evaluate the effect of using different numbers of representations/modalities in decoding. We compare our MMPI using both representations in an alternating manner with a variant using LiDAR representation in all decoder layers. It is observed that using both representations is beneficial, verifying our design consideration.

**Number of decoder layers** As shown in Table 3(c), increasing the number of decoder layers up to 5 layers can consistently improve the performance whilst introducing negligible latency. LiDAR-only denotes the Transfusion-L [1] baseline.

**Number of queries** Since our query embeddings are initialized in a non-parametric and input-dependent manner as in [1], the number of queries are adjustable during inference. In Table 3(d), we evaluate different combinations of query numbers for training and test. Overall, the performance is stable over different choices with 200/300 for training/test as the best setting.

### 4.5 Ablations on the encoder

**Multi-modal representational interaction vs. fusion** To precisely demonstrate the superiority of our multi-modal representational interaction, we compare a naive version of our `DeepInteraction` with conventional representational fusion strategy as presented in Transfusion [1]. We limit our `DeepInteraction` using the same number of encoder and decoder layers as [1] for a fair comparison. Table 4(c) shows that our representational interaction is clearly more effective.

**Encoder design** We ablate the design of our encoder with focus on multi-modal representational interaction (MMRI) and intra-modal representational learning (IML). We have a couple of observations from Table 4(a): (1) Our MMRI can significantly improve the performance over IML; (2) MMRI and IML can work well together for further performance gain. As seen from Table 4(b), stacking our encoder layers for iterative MMRI is beneficial.

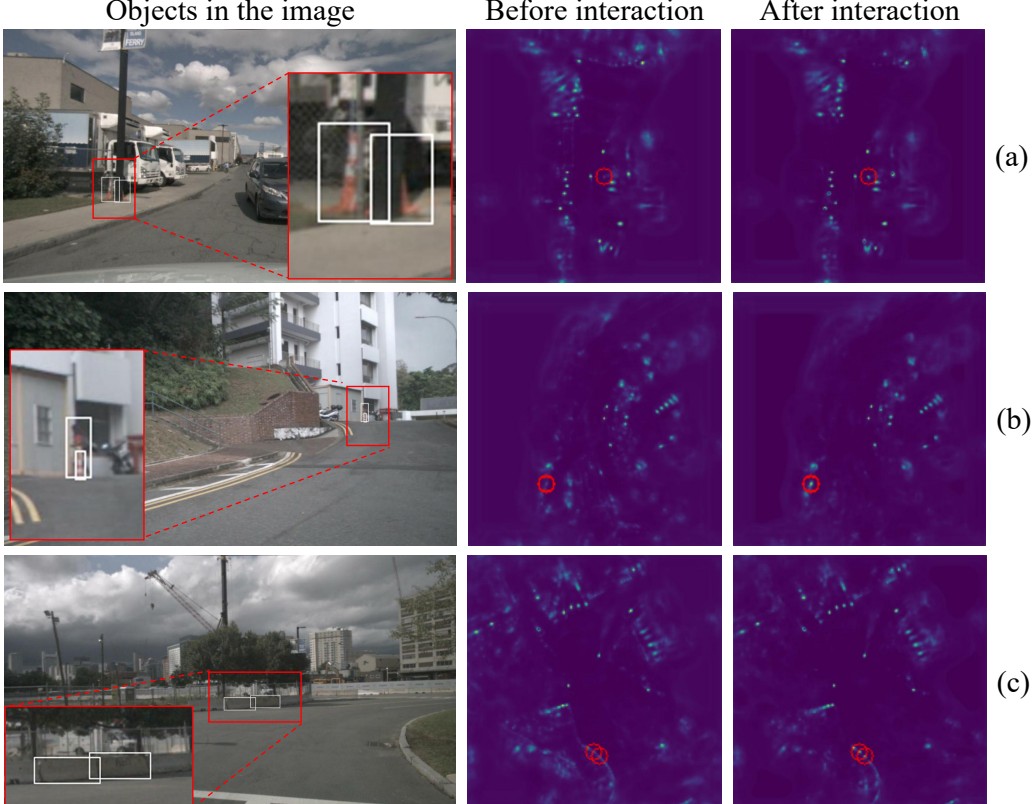

|  | Objects in the image | Before interaction | After interaction |
| --- | --- | --- | --- |

Figure 5: Illustrations of the heatmaps predicted from BEV representations *before and after* representational interactions. All samples are from the nuScenes `val` split. **(a)** A case of the occluded tiny object. **(b)** A case of small object at long distance. **(c)** An example of two adjacent barriers which are connected together in LiDAR point clouds and thus it is difficult to have their instance-level understanding without the help of visual clues.

**Qualitative results of representational interaction** To gain more insight about the effect of our multi-modal representational interaction (MMRI), we visualize the heatmaps of challenging cases. We observe from Figure 5 that without the assistance of MMRI, some objects cannot be detected when using LiDAR only (the middle column). The locations of these objects are highlighted by red circles in the heatmap and white bounding boxes in the RGB image below.

Table 4: Ablation studies on the representational interaction encoder. The mAP and NDS are evaluated on the nuScenes `val` set.

| IML | MMRI | mAP | NDS |
| --- | --- | --- | --- |
| ✓ |  | 68.1 | 71.9 |
|  | ✓ | 69.5 | 72.5 |
| ✓ | ✓ | **69.9** | **72.6** |

(a) Encoder design. `IML`: Intra-modal learning; `MMRI`: Multi-modal representational interaction.

| # of encoder layers | mAP | NDS |
| --- | --- | --- |
| w/o | 66.4 | 70.7 |
| 1 | 67.7 | 71.2 |
| 2 | **69.9** | **72.6** |

(b) The number of encoder layers.

| Method | mAP | NDS |
| --- | --- | --- |
| Representational fusion | 67.5 | 71.3 |
| Representational interaction (Ours) | **68.7** | **71.9** |

(c) Our representational interaction vs. conventional representational fusion (*e.g.*, Transfusion [1]).

Table 5: Evaluation on different LiDAR backbones. The mAP and NDS are evaluated on the nuScenes `val` set.

| Methods | Modality | mAP | NDS |
|---|---|---|---|
| PointPillars [22] | L | 46.2 | 59.1 |
| +Transfusion-L [1] | L | 54.5 | 62.7 |
| +Transfusion [1] | L+C | 58.3 | 64.5 |
| +DeepInteraction | L+C | **60.0** | **65.6** |

(a) Comparison between pillar-based methods.

| Methods | Modality | mAP | NDS |
|---|---|---|---|
| VoxelNet [47] | L | 52.6 | 63.0 |
| +Transfusion-L [1] | L | 65.1 | 70.1 |
| +Transfusion [1] | L+C | 67.5 | 71.3 |
| +DeepInteraction | L+C | **69.9** | **72.6** |

(b) Comparison between voxel-based methods.

Table 6: Comparison with the LiDAR-only baseline Transfusion-L [1] on nuScenes `val` split.

| Method | mAP | NDS | Car | Truck | C.V. | Bus | T.L. | B.R. | M.T. | Bike | Ped. | T.C. |
|---|---|---|---|---|---|---|---|---|---|---|---|---|
| Transfusion-L [1] | 65.1 | 70.1 | 86.5 | 59.6 | 25.4 | 74.4 | 42.2 | 74.1 | 72.1 | 56.0 | 86.6 | 74.1 |
| **DeepInteraction** | **69.9** | **72.6** | **88.5** | **64.4** | **30.1** | **79.2** | **44.6** | **76.4** | **79.0** | **67.8** | **88.9** | **80.0** |

Concretely, the sample (a) suggests that camera information is helpful to recover partially obscured tiny objects with sparse observation in the point cloud. The sample (b) shows a representative case where some distant objects can be recognized successfully due to the help of visual information. From the sample (c), we observe that the centers of some barriers yield a more distinct activation in the heatmap after representational interaction. This is probably due to that it is too difficult to locate the boundaries of several consecutive barriers from LiDAR point clouds only.

### 4.6 Ablation on LiDAR backbones

We examine the generality of our framework with two different LiDAR backbones: PointPillars [22] and VoxelNet [47]. For PointPillars, we set the voxel size to (0.2m, 0.2m) while keeping the remaining settings same as DeepInteraction-base. For fair comparison, we use the same number of queries as TransFusion [1]. As shown in Table 5, due to the proposed multi-modal interaction strategy, `DeepInteraction` exhibits consistent improvements over LiDAR-only baseline using either backbone (by 5.5% mAP for voxel-based backbone, and 4.4% mAP for pillar-based backbone). This manifests the generality of our `DeepInteraction` across varying point cloud encoder.

### 4.7 Performance breakdown

To demonstrate more fine-grained performance analysis, we compare our `DeepInteraction` and our LiDAR-only baseline Transfusion [1] at the category level in terms of mAP on nuScenes `val` set. We can see from Table 6 that our fusion approach achieves remarkable improvements on all the categories, especially on tiny or rare object categories (+11.8% mAP for bicycle, +6.9% mAP for motorcycle, and +5.9% mAP for traffic cone).

## 5 Conclusion

In this work, we have presented a novel 3D object detection method `DeepInteraction` for exploring the intrinsic multi-modal complementary nature. This key idea is to maintain two modality-specific representations and establish interactions between them for both representation learning and pre-dictive decoding. This strategy is designed particularly to resolve the fundamental limitation of existing unilateral fusion approaches that image representation are insufficiently exploited due to their auxiliary-source role treatment. Extensive experiments demonstrate our proposed `DeepInteraction` yields new state of the art on the nuScenes benchmark dataset and achieves the first position at the highly competitive nuScenes 3D object detection leaderboard.

**Acknowledgement**    This work was supported in part by National Natural Science Foundation of China (Grant No. 6210020439), Lingang Laboratory (Grant No. LG-QS-202202-07), Natural Science Foundation of Shanghai (Grant No. 22ZR1407500), Science and Technology Innovation 2030 - Brain Science and Brain-Inspired Intelligence Project (Grant No. 2021ZD0200204) and Shanghai Municipal Science and Technology Major Project (Grant No. 2018SHZDZX01).

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
