# A Appendix

## A.1 Performance breakdown for categories

In table 1, we report the performance on each category. It is shown that our `DeepInteraction` performs the best among all the competitors across the most of object categories.

Table 1: Comparison with state-of-the-art methods on the nuScenes `test` set. `Metrics:` mAP(%)↑, NDS(%)↑, and AP(%)↑ for each category. 'C.V.', 'Ped.', and 'T.C.','M.T.' and 'T.L.' are short for construction vehicle, pedestrian, traffic cone, motor and trailer respectively. 'L' and 'C' represent LiDAR and camera, respectively. † denotes test-time augmentation is used. § denotes that test-time augmentation and model ensemble both are applied for testing.

| Method | Mod. | mAP | NDS | Car | Truck | C.V. | Bus | T.L. | B.R. | M.T. | Bike | Ped. | T.C. |
|---|---|---|---|---|---|---|---|---|---|---|---|---|---|
| CenterPoint [9]† | L | 60.3 | 67.3 | 85.2 | 53.5 | 20.0 | 63.6 | 56.0 | 71.1 | 59.5 | 30.7 | 84.6 | 78.4 |
| Focals Conv [2] | L | 63.8 | 70.0 | 86.7 | 56.3 | 23.8 | 67.7 | 59.5 | 74.1 | 64.5 | 36.3 | 87.5 | 81.4 |
| TransFusion-L [1] | L | 65.5 | 70.2 | 86.2 | 56.7 | 28.2 | 66.3 | 58.8 | 78.2 | 68.3 | 44.2 | 86.1 | 82.0 |
| LargeKernel [3] | L | 65.3 | 70.5 | 85.9 | 55.3 | 26.8 | 66.2 | 60.2 | 74.3 | 72.5 | 46.6 | 85.6 | 80.0 |
| PointAug. [7]† | L+C | 66.8 | 71.0 | 87.5 | 57.3 | 28.0 | 65.2 | 60.7 | 72.6 | 74.3 | 50.9 | 87.9 | 83.6 |
| MVP [10] | L+C | 66.4 | 70.5 | 86.8 | 58.5 | 26.1 | 67.4 | 57.3 | 74.8 | 70.0 | 49.3 | 89.1 | 85.0 |
| FusionPainting [8] | L+C | 68.1 | 71.6 | 87.1 | 60.8 | 30.0 | 68.5 | 61.7 | 71.8 | 74.7 | 53.5 | 88.3 | 85.0 |
| AutoAlign [4] | L+C | 68.4 | 72.4 | 87.0 | 59.0 | 33.1 | 69.3 | 59.3 | 78.0 | 72.9 | 52.1 | 87.6 | 85.1 |
| FUTR3D [4] | L+C | 68.4 | 72.4 | 87.0 | 59.0 | 33.1 | 69.3 | 59.3 | 78.0 | 72.9 | 52.1 | 87.6 | 85.1 |
| TransFusion [1] | L+C | 68.9 | 71.7 | 87.1 | 60.0 | 33.1 | 68.3 | 60.8 | 78.1 | 73.6 | 52.9 | 88.4 | 86.7 |
| BEVFusion [5] | L+C | 69.2 | 71.8 | 88.1 | 60.9 | 34.4 | 69.3 | 62.1 | 78.2 | 72.2 | 52.2 | 89.2 | 85.5 |
| BEVFusion [6] | L+C | 70.2 | 72.9 | **88.6** | 60.1 | **39.3** | 69.8 | **63.8** | 80.0 | 74.1 | 51.0 | 89.2 | 86.5 |
| **DeepInteraction-base** | L+C | **70.8** | **73.4** | 87.9 | 60.2 | 37.5 | **70.8** | **63.8** | 80.4 | 75.4 | 54.5 | **91.7** | **87.2** |
| Focals Conv-F [2]† | L+C | 70.1 | 73.6 | 87.5 | 60.0 | 32.6 | 69.9 | 64.0 | 71.8 | 81.1 | 59.2 | 89.0 | 85.5 |
| LargeKernel3D-F [3]† | L+C | 71.1 | 74.2 | 88.1 | 60.3 | 34.3 | 69.1 | **66.5** | 75.5 | 82.0 | 60.3 | 89.6 | 85.7 |
| **DeepInteraction-large†** | L+C | **74.1** | **75.5** | 88.8 | 64.0 | 40.8 | 70.9 | 62.7 | 82.3 | 85.3 | 64.5 | 92.6 | 89.3 |
| BEVFusion-e [6]§ | L+C | 75.0 | 76.1 | **90.5** | **65.8** | 42.6 | **74.2** | **67.4** | 81.1 | 84.4 | 62.9 | 91.8 | 89.4 |
| **DeepInteraction-e§** | L+C | **75.7** | **76.3** | 89.0 | 64.5 | **44.7** | 74.2 | 66.0 | **83.5** | **85.4** | **66.4** | **92.8** | **90.9** |

## A.2 Discussions of potential societal impacts

Fusing multi-modal information allows to compensate for the shortcomings of single modality in 3D object detection, leading to more more accurate and robust performance. In practice, a stronger 3D object detection method as our `DeepInteraction` model is expected to reduce the potential accidents of self-driving cars. This improves the safety and reliability of autonomous driving. However, multi-modal algorithms often require more powerful computing devices and run at a higher cost. This raises a need for improving the system efficiency to be resolved in the future.

## A.3 Limitations

All the components for multi-modal fusion in our `DeepInteraction` have no preference to any per-modal representations. However, the initial queries are derived from LiDAR BEV, albeit fused with image features. We will explore how to generate initial queries from both modalities (*i.e.*, LiDAR's bird-eyes-view and camera's front-view).

Our method involves explicit 2D-3D mapping, hence is conditioned on the calibration quality of the sensors. To relax this condition, a potential method is to exploit the attention mechanism to allow the network to automatically establish alignment between multi-modal features.

Finally, our current model design does not take into account model efficiency. In the future, we will develop a more advanced framework which can adaptively select more cost-effective combinations of interaction operators in order to optimize the trade-off between performance, efficiency and robustness.

## A.4 More visualizations

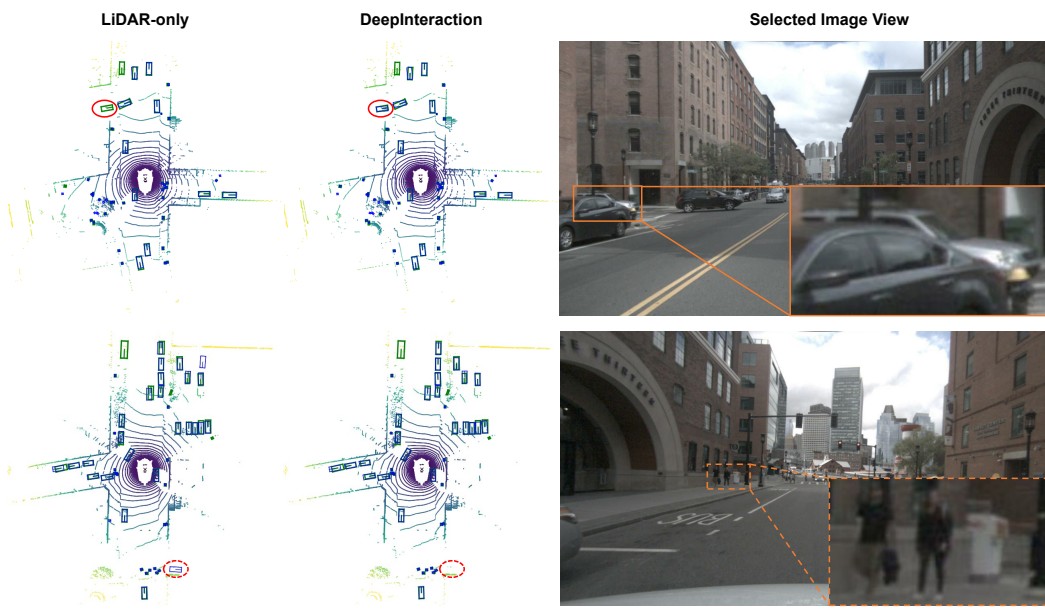

Figure 1: Qualitative comparison between LiDAR-only baseline and our `DeepInteraction`. Blue boxes and green boxes are the predictions and ground-truth, respectively. Solid eclipse indicates false negative, and dashed eclipse indicates false positive.

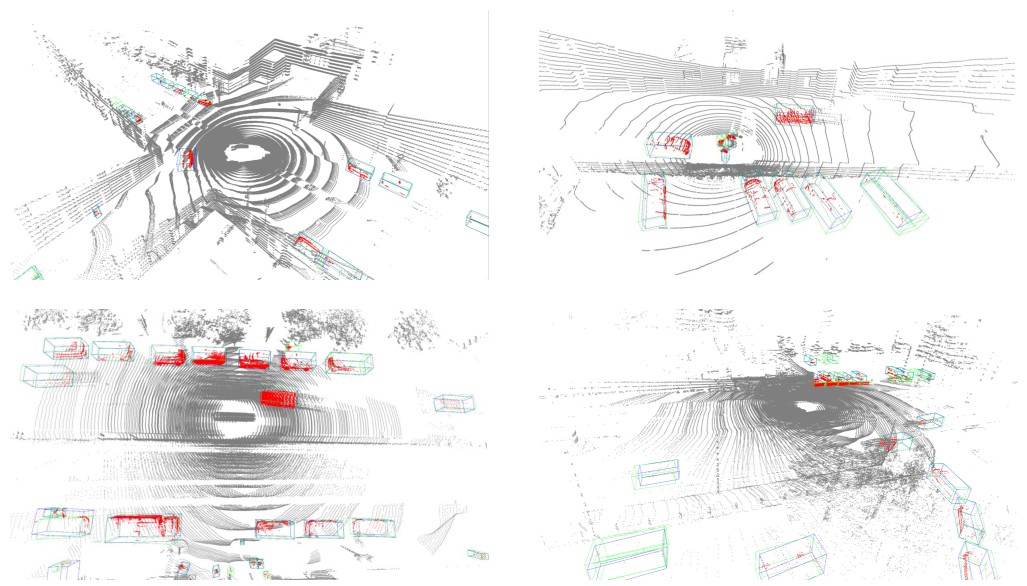

Figure 2: Detection results shown in point clouds by `DeepInteraction` on nuScenes `val` set. The bounding boxes of ground-truth and predictions are in the color blue and green respectively. Best viewed on screen.