# OpenReview forum: "DeepInteraction: 3D Object Detection via Modality Interaction"
_NeurIPS.cc/2022/Conference — NeurIPS 2022 Accept_

### Official Review · Reviewer_83YX · 2022-07-09

**Rating:** 7
**Confidence:** 4
**Soundness:** 3 good
**Presentation:** 3 good
**Contribution:** 3 good

**Summary:**

(1) The paper reveals and gives an analysis of the unilateral limitation of feature aggregation on different modalities for existing methods.
(2) The paper proposes a new framework for bilateral feature interaction and association on different modalities.
(3) The paper achieves state-of-the-art results on nuScenes benchmark.

**Questions:**

Authors are encouraged to provide additional experiments for the two concerns I list in the weakness part:

(1) The comparison of speed and memory.
(2) Some visual results(e.g., heatmap of two sources) to convince the effect of the proposed bilateral framework.

**Limitations:**

The authors have presented the potential societal impacts and limitations in their supplementary material. Overall, they look good to me.

**Strengths And Weaknesses:**

Strengths:
(1) The paper is well-written and easy to follow, especially for demonstrating its motivation in the introduction session.
(2) The experiments are sufficient and convincing for both comparing with previous methods and ablation studies.

Weaknesses:
(1) The paper lacks the comparison of speed and memory between different methods. For the autonomous driving scenarios whether the model can run in real-time is also important. I'm curious about whether the DETR-based framework can run as fast as, at least not slower too much compared with previous SOTAs.
(2) Although the proposed bilateral system is reasonable, it would be great to show some visualizations of the heat map of the features from the two modalities to help readers get a deeper understanding on how the visual information from the two sources is well selected and utilized.

---

> ### Author Response · Authors · 2022-08-02
> **Response to Reviewer 83YX**
>
> We thank the reviewer for the detailed review as well as the suggestions for improvement. Our response to the reviewer’s comments is below:
>
> **Q1: The comparison of speed and model size.**
>
> Thanks for this valuable suggestion.
>
> We have presented the latency of DeepInteraction in Table 3d in the main text. As suggested, we have further compared the inference latency and model size between our DeepInteraction and a number of representative multimodal 3D detection methods. This test was conducted on a Tesla V100 GPU. The latency and parameter number of MVP [2] are obtained with its virtual point generation algorithm and object detection algorithm considered. The mAP and NDS of PointAugmentation [1] and MVP [2] are taken from the original papers. We tested latency, FPS, and parameter numbers using their open source code and their published configuration file. The table below shows that our method runs the fastest among the multimodal fusion methods in the table below. The same is observed for the number of parameters. Please note in this work we do not focus on the model efficiency.
>
> Table 1: The comparison of speed and model size.
> | Method |mAP(%)↑|NDS(%)↑|Latency(ms)↓|Pamras(M)↓|FPS↑|
> |:----|:----:|:----:|:----:|:----:|:----:|
> |PointAugmenting [1]|66.8|71.0|455|**28.2**|2.2|
> |MVP [2]|66.4|70.5|830|124.2|1.2|
> |DeepInteraction|**70.8**|**73.4**|**357**|59.4|**2.8**|
>
> **Q2: More visualizations.**
>
> Great suggestion. We have now provided extra visualization examples in Figure 3 of our revised supplementary material. In particular, with multimodal fusion by our DeepInteraction, the model can successfully identify those objects that are difficult to recall in the LiDAR point cloud only.
>
>
> >[1] Wang, Chunwei, et al. Pointaugmenting: Cross-modal augmentation for 3d object detection. *CVPR* 2021.
>
> >[2] Yin, et al. Multimodal virtual point 3d detection. *NeurIPS* 2021.

---

### Official Review · Reviewer_F1dJ · 2022-07-11

**Rating:** 1
**Confidence:** 4
**Ethics Flag:** Yes
**Soundness:** 3 good
**Presentation:** 3 good
**Contribution:** 2 fair

**Summary:**

This work tackles the task of 3D object detection in autonomous driving scenarios. The claimed key contribution is the proper combination of two input modalities: RGB camera images and point clouds from LiDAR scanners (unlike prior methods which treat one of the modalities as auxiliary). The proposed method outperforms prior methods on the nuScenes benchmark. The method is analyzed on the same dataset.

**Questions:**

(1) Which experiment in the ablation study shows that the bilateral feature fusion is indeed the key component necessary for the improved scores?

(2) How does the model perform on other datasets e.g. KITTI 3D object detection?

**Minor Details**

- Fig.2: inconsistent notation \phi <-> \varphi in (a) and (b).
- l.59 ‘preform’ → ‘perform’
- reference [5] and [6] are the same


**Ethics Review Area:**

["Research Integrity Issues (e.g., plagiarism)", "Responsible Research Practice (e.g., IRB, documentation, research ethics)"]

**Limitations:**

Potential negative societal impact is not discussed (there is a section in the supplementary titled as such but discusses only research impact). Limitations are only discussed on an abstract meta level. Overall, the two sections (E, F) in the supplementary are not adding any real value to the paper and seem to exist only to tick the required boxes.

**Strengths And Weaknesses:**

- The paper makes a strong point that the bilateral fusion between camera and lidar modalities is the key part that contributes to the strong performance of the proposed model. From the ablation study alone it is not directly clear if the improvements come from the change from unilateral to bilateral modality information flow or from stronger backbones which seem to have the biggest impact according to Table (e).

- The method is evaluated only on a single dataset. There are multiple other popular 3D object detection benchmarks, e.g., [KITTI](http://www.cvlibs.net/datasets/kitti/eval_object.php?obj_benchmark=3d).
Evaluation on more datasets would show the generalization capability of the model.

- Important related work on image depth fusion and transformer-based 3D object detection appears to be missing, e.g., [ImVoteNet](https://arxiv.org/pdf/2001.10692.pdf), [3DETR](https://arxiv.org/abs/2109.08141) [Misra et al. ICCV 2021].

---

> ### Author Response · Authors · 2022-08-02
> **Response to Reviewer F1dJ**
>
> We thank the reviewer for the detailed review as well as the suggestions for improvement. Our response to the reviewer’s comments is below:
>
> **Q1: Ablation on the bilateral feature fusion *vs* unilateral fusion.**
>
> As clearly presented in Table 3c of the main paper, the bilateral modality interaction we introduce in this paper is a key performance contributor. Specifically, the first row of Table 3c gives the result (only 66.4% mAP) **without** the bilateral interaction encoder. This is clearly inferior to the variants (other rows in Table 3c) equipped with our proposed bilateral interaction encoders with various layers.
>
> To more precisely demonstrate the superiority of bilateral feature interaction, we have now compared our bilateral interaction (naive version of our DeepInteraction) with the classical unilateral fusion alternative Transfusion [1]. Here we limit our DeepInteraction using the same number of encoder layers as Transfusion for a fair comparison. The table below shows that our bilateral interaction is clearly more effective for modality fusion.
>
> Table 1: Ablation on the bilateral feature fusion *vs* unilateral fusion.
> |Method|mAP|NDS|
> |:----|:----:|:----:|
> |Transfusion w/ unilateral|67.5|71.3|
> |Transfusion w/ bilateral|**68.7**|**71.9**|
>
> **Q2: The improvements come from the stronger backbone which seems to have the biggest impact according to Table (e)?**
>
> No.
>
> **First**, we adopt the widely used Resnet-50 and Voxlnet as our image and Lidar backbone without bells and whistles.
>
> **Second**, as clearly presented in Lines 237-244 and Table 3e, with the same backbone, our DeepInteraction is clearly superior to the previous state-of-the-art method Transfusion [1] and this advantage is generic to the backbone selected (*e.g.*, PointPillars or VoxelNet). It is worthy to note that backbone is orthogonal to our model design novelty (*e.g.*, modality interaction) and not comparable in an apple-to-apple manner (a very basic principle).
>
> **Third**, our DeepInteraction with a ResNet-50 backbone can outperform the latest concurrent work BEVFusion [3] with the stronger Tiny-Swin backbone on the nuScenes dataset, further demonstrating the advantage of our modality fusion strategy rather than the backbone.
>
> **Q3:Results on other datasets?**
>
> As suggested, during the rebuttal period, we have further evaluated our DeepInteraction on the Waymo and KITTI benchmark. Due to limited time and computational resources, we can not well tune the hyperparameters for optimal performance.
>
> (1) Waymo open dataset
>
> For the Waymo Open dataset, we used the Transfusion-L [1] trained on Waymo as our LiDAR-only baseline and used the ResNet-50 from the cascade mask RCNN pretrained on the nuImage instance segmentation task as our image backbone (same as on nuScenes in the main paper).
> The L2 mAPH of our approach are listed below.
>
> Table 2: LEVEL_2 APH of the Waymo validation set (%).
> |Method|APH/L2@Vehicle|APH/L2@Pedestrian|
> |:----|:----:|:----:|
> |PointAugmenting [2]|62.2|64.6|
> |Transfusion [1]|65.1|64.0|
> |DeepInteraction|**65.4**|**64.9**|
>
> From this table, we can see that our model can achieve superior performance over the latest alternative approaches, with the biggest gain in the pedestrian category with small size.
>
> (2) KITTI dataset
>
> The experiment on the KITTI dataset follows the same recipe as on Waymo. The table below presents the comparison of 3D AP on the moderate split of the KITTI validation set. Beyond the LiDAR-only baseline Transfusion-L, we also reproduced Transfusion-LC on KITTI for comparison. We observe from the table below that our method can achieve the best results for both car and pedestrian. This demonstrates that the proposed fusion strategy can consistently bring in benefits for LiDAR with more beams.
>
> Table 3: The performance on the KITTI validation set (%).
> |Method|3D AP@Vehicle|3D AP@Pedestian|
> |:----|:----:|:----:|
> |Transfusion-L  [1]| 69.8 |51.9|
> |Transfusion-LC [1]|70.0|52.6|
> |DeepInteraction|**70.2**|**53.5**|
>
> **Q4: Missing related work.**
>
> Thanks. The related work of our first submission mainly focuses on previous works for outdoor scenarios. We have now added the suggested work although with less relevance in the revised paper.
>
> **Q5: Typos**
>
> Thanks, we have corrected them in the revised paper.
>
> **Q6: Additional discussion of social impacts and limitations**
>
> Thanks. We have added more detailed discussion on the limitations and potential impacts in the revised supplementary.
>
> >[1] Bai, Xuyang, et al. Transfusion: Robust lidar-camera fusion for 3d object detection with transformers. *CVPR* 2022.
>
> >[2] Chunwei Wang, et al. PointAugmenting: Cross-modal augmentation for 3d object detection. *CVPR* 2021.
>
> >[3] BEVFusion: Multi-Task Multi-Sensor Fusion with Unified Bird's-Eye View Representation, 2022

---

### Official Review · Reviewer_2uMx · 2022-07-12

**Rating:** 6
**Confidence:** 3
**Soundness:** 2 fair
**Presentation:** 4 excellent
**Contribution:** 3 good

**Summary:**

The paper introduces DeepInteraction, a new approach to fuse rgb and depth input to do 3D object detection. Prior works only add rgb features to the Lidar point cloud, while DeepInteraction treat them equally and fuse them in the transformer layers. Experiments are conducted on nuScenes benchmark and DeepInteraction achieves state-of-the-art performance.


**Questions:**

See weaknesses.

**Limitations:**

Limitations are discussed in the paper.

**Strengths And Weaknesses:**

Strengths:

- DeepInteraction achieves state-of-art performance on nuScenes benchmark. Under the same evaluation setting (without ensemble), it generally improves mAP by 2 points whether test-time augmentation is used or not.
- The experiments on nuScenes are extensive. It's interesting to see DeepInteraction has reasonable performance whether the 3D backbone is PointPillars or VoxelNet, and VoxelNet works better.

Weaknesses:

- How does the approach work on other automonous driving datasets? I'm convinced DeepInteraction helps recover tiny objects which are just several pixels in the Lidar points. But are such cases really general in other benchmarks? Or, are there any reasons it cannot be run on other benchmarks?

---

> ### Author Response · Authors · 2022-08-02
> **Response to Reviewer 2uMx**
>
> We thank the reviewer for the detailed review as well as the suggestions for improvement. Our response to the reviewer’s comments is below:
>
> **Q1: Results on the other datasets.**
>
> Thanks.  As suggested, we have further evaluated our DeepInteraction on the Waymo benchmark. Due to limited time and computational resources, we have not exhaustively tuned the hyperparameters.
> We used the Transfusion-L [1] trained on Waymo as our LiDAR-only baseline and used the ResNet-50 from the cascade mask RCNN pretrained on the nuImage instance segmentation task as our image backbone (same as on nuScenes in the main paper).
>
> Table 1: The LEVEL_2 APH results on the Waymo validation set.
>
> | Method | APH/L2@Vehicle| APH/L2@Pedestrian|
> |:----|:----:|:----:|
> |PointAugmenting [2]|62.2|64.6|
> |Transfusion [1]|65.1|64.0|
> |DeepInteraction|**65.4**|**64.9**|
>
> From this table, we can see that our model can achieve slightly superior performance compared to the latest alternative approaches, with the biggest gain on the category with small size (*e.g.*, pedestrian).
>
> **Q2: Generality of tiny objects containing only several Lidar points.**
>
> Good question. To understand whether tiny objects with a few pixels in the Lidar points are general in other datasets, we present statistics of the number of points within ground truth bounding boxes in the validation split of each dataset.
>
> Table 2: The frequency (%) of objects with different numbers of LiDAR data.
> |Number of LiDAR point|0~20|20~40|40~60|60~80|80+|
> |:----|:----:|:----:|:----:|:----:|:----:|
> |nuScenes|69.3|10.1|4.4|2.7|13.5|
> |Waymo|24.1|14.5|9.0|6.3|46.1|
>
> From the above table, we can observe that tiny objects (*e.g.*, < 20 points) are highly general across different scenarios/benchmarks at varying degrees. Specifically, there are 69.3% and 24.1% of objects containing less than 20 points in nuScenes and Waymo respectively.
>
>
> >[1] Bai, Xuyang, et al. Transfusion: Robust lidar-camera fusion for 3d object detection with transformers. *CVPR* 2022.
>
> >[2] Chunwei Wang, et al. PointAugmenting: Cross-modal augmentation for 3d object detection. *CVPR* 2021.

---

> > ### Comment · Reviewer_2uMx · 2022-08-08
> > **Re: Response to Reviewer 2uMx**
> >
> > Thanks for your updates! After reading other reviews, I'm happy to keep my current rating.

---

### Review · Ethics_Reviewer_4TUF · 2022-08-05

**Recommendation:**

The authors might consider including information about potential impact (improvement?) to safety in the supplementary materials.


**Ethics Review:**

A reviewer noted “Research Integrity Issues (e.g., plagiarism), Responsible Research Practice (e.g., IRB, documentation, research ethics)”. However, they did not provide additional information in their comment. Another reviewer noted: “Potential negative societal impact is not discussed (there is a section in the supplementary titled as such but discusses only research impact).”

---

### Review · Ethics_Reviewer_oS3P · 2022-08-24

**Recommendation:**

In light of these questions and my own understanding of the situation, I'm inclined to say that there is no double-blind reviewing policy violation, but I am curious about the thoughts from the other reviewers. I particularly want to hear about how other researchers in the field operate with these anonymity constraints as well as how conferences (including NeurIPS!) have set regulations around this.

**Ethics Review:**

The paper describes a 3D object detection architecture for 3D LiDAR point clouds. The question at hand is whether or not the authors violated double-blind reviewing.

Here are the facts:
 - The authors reference their position and test set results on a public leaderboard.
 -The leaderboard is the nuScenes detection task leaderboard [1], where the authors are de-identified. The entry is date-stamped the day before the NeurIPS deadline.
 - The relevant sections of the leaderboard are reproduced and de-identified in the paper. The website link for the leaderboard is not linked anywhere in the paper.
 - Reviewer F1dJ has raised the issue and provided the link in question

Because I am not familiar with this particular subfield, I have the following questions:

1. How do other teams in this field and on these kinds of tasks handle the de-identification? By visual inspection 40-50% of the leaderboard entries are anonymized. Can teams submit results anonymously and then de-identify after the review period is over?

2. Does NeurIPS have an official policy for authors to not post on leaderboards and mention them in the paper directly? Would it be different if the authors had merely described "an un-named detection task" and provided test-set results against several known baselines?

3. Do the NeurIPS reviewers receive instructions to not knowingly seek out and try to de-identify the authors? Again, the leaderboard is not linked in the paper anywhere. The relevant portions of the leaderboard are reproduced in Table 1 in the paper. I can see an argument that this is similar to posting a Github repo in advance where the NeurIPS reviewer could find the Github repo with names if they searched for the exact paper title, but they may be instructed not to. Alternatively, the conference could instruct authors not to post ANY identifiable information on the internet.

In light of these questions and my own understanding of the situation, I'm inclined to say that there is no double-blind reviewing policy violation, but I am curious about the thoughts from the other reviewers. I particularly want to hear about how other researchers in the field operate with these anonymity constraints as well as how conferences (including NeurIPS!) have set regulations around this.

[1] https://www.nuscenes.org/object-detection?externalData=all&mapData=all&modalities=Any

---

### Meta-Review · Program_Chairs · 2022-08-24

**Recommendation:** Accept
**Confidence:** Certain

**Metareview:**

This work was overall positively technically evaluated with some concerns mainly related to limited experimental validation, the need to some additional justifications and explanation, and the missing computational cost analysis.
The provided rebuttal responded sufficiently well to these concerns and the overall evaluation is positive.



**Award:**

No

---

### Decision · Program_Chairs · 2022-09-14

Accept